# Feasibility of Ultrasound-Guided Axillary Vein Puncture under Valsalva Maneuver for Diagnostic and Cardiovascular Interventional Purposes: Pacemaker and Cardioverter-Defibrillator Implantation

**DOI:** 10.3390/diagnostics13203274

**Published:** 2023-10-21

**Authors:** Biagio Sassone, Enrico Bertagnin, Santo Virzì, Giuseppe Simeti, Paolo Tolomeo

**Affiliations:** 1Department of Translational Medicine, University of Ferrara, 44121 Ferrara, Italy; 2Cardiothoracic Vascular Department, Division of Provincial Cardiology, Santissima Annunziata Hospital and Delta Hospital, Azienda Unità Sanitaria Locale di Ferrara, 44042 Ferrara, Italy; enrico.bertagnin@ausl.fe.it (E.B.); s.virzi@ausl.fe.it (S.V.); giuseppesimeti89@gmail.com (G.S.); paolo.tolomeo@ausl.fe.it (P.T.)

**Keywords:** Valsalva maneuver, ultrasound, axillary vein, pacemaker

## Abstract

Although ultrasound-guided axillary vein access (USGAVA) has proven to be a highly effective and safe method for cardiac electronic implantable device (CIED) lead placement, the collapsibility of the axillary vein (AV) during tidal breathing can lead to narrowing or complete collapse, posing a challenge for successful vein puncture and cannulation. We investigated the potential of the Valsalva maneuver (Vm) as a facilitating technique for USGAVA in this context. Out of 148 patients undergoing CIED implantation via USGAVA, 41 were asked to perform the Vm, because they were considered unsuitable for venipuncture due to a narrower AV diameter, as assessed by ultrasound (2.7 ± 1.7 mm vs. 9.1 ± 3.3 mm, *p* < 0.0001). Among them, 37 patients were able to perform the Vm correctly. Overall, the Vm resulted in an average increase in the AV diameter of 4.9 ± 3.4 mm (*p* < 0.001). USGAVA performed during the Vm was successful in 30 patients (81%), and no Vm-related complications were observed during the 30-day follow-up. In patients with unsuccessful USGAVA, the Vm resulted in a notably smaller increase in AV diameter (0.5 ± 0.3 mm vs. 6.0 ± 2.8 mm, *p* < 0.0001) compared to patients who achieved successful USGAVA, while performing the Vm. Therefore, the Vm is a feasible maneuver to enhance AV diameter and the success rate of USGAVA in most patients undergoing CIED implantation while maintaining safety.

## 1. Introduction

For over 60 years, either the cephalic vein or the subclavian vein have been the preferred routes for transvenous cardiovascular implantable electronic device (CIED) leads implantation, including pacemakers and cardioverter-defibrillators [1]. While CIED implantation is generally regarded as a safe procedure [2], it is noteworthy that intrathoracic puncture of the subclavian vein carries the potential risk of life-threatening hazards, including hemothorax and tension pneumothorax [3,4,5]. Additionally, there are possible serious long-term complications such as lead crush [6,7,8,9]. Conversely, the cephalic vein cutdown is a highly safe approach, as there is no need for a central venipuncture [10,11,12,13]. Nonetheless, its anatomical characteristics (i.e., thinness and tortuosity) lower the success rate of cannulation, especially when managing the placement of multiple leads [14,15,16]. In the last decade, the puncture of the infraclavicular axillary vein (AV) has been emerging as an alternative technique for CIED leads insertion [17,18]. This venous approach is characterized by a higher success rate and greater safety as compared to cephalic vein cutdown and subclavian vein puncture, respectively [19,20,21]. The main advantages of the AV over traditional venous accesses are represented by its linear course and large diameter, enabling easy insertion of multiple leads [22]. Moreover, its entirely extrathoracic course enables puncture outside the rib cage, thereby reducing the risk of severe complications associated with central intrathoracic venipuncture [23,24].

While most operators perform AV puncture guided by fluoroscopy landmarks or contrast venography [25,26,27], ultrasound-guided AV access (USGAVA) has been gaining widespread acceptance among operators in more recent times [28]. Thanks to the direct visualization of the targeted vein without the use of nephrotoxic iodinated contrast media, USGAVA minimizes exposure to ionizing radiation during fluoroscopy [29,30]. Despite USGAVA proving to be a reliable method for CIED implantation, the respiratory variation in AV size during tidal breathing, which can lead to complete venous collapse [31,32], represents a significant challenge for successful vein puncture and cannulation. Asking the patient to hold a forced expiration for a few seconds, as prescribed by the Valsalva maneuver, has the potential to address these challenges. The Valsalva maneuver is recognized as a technique capable of expanding the AV dimensions and counteracting respiratory collapse [33,34]. Nevertheless, data regarding the use of such maneuver to facilitate USGAVA during CIED implantation are currently lacking.

In this prospective observational study, we share our firsthand experience of integrating the Valsalva maneuver within the USGAVA procedure to facilitate pacemaker and cardioverter-defibrillator lead implantation in patients initially considered unsuitable for AV puncture, due to its small size and high collapsibility. We specifically focused on assessing the feasibility, effectiveness, and safety of employing the Valsalva maneuver in this context.

## 2. Materials and Methods

### 2.1. Setting and Ethics

The present study was conducted at Division of Provincial Cardiology of Ferrara, Italy, and was approved by the local Ethics Committee of Area Vasta Emilia Centro (identifier: 759/2021/Oss/AUSLFe). It conformed to the principles of the Declaration of Helsinki. Written informed consent was obtained from all eligible patients before the implantation procedure.

### 2.2. Patient Selection

The study enrolled, from June 2021 to June 2023, all consecutive adult (>18 years) patients referred to our tertiary cardiology center for pacemaker and cardioverter-defibrillator implantation, also including devices for cardiac resynchronization therapy, who systematically underwent USGAVA as the initial approach for leads’ insertion into the central venous system through the AV. Lead revisions and device upgrades were excluded from the study. USGAVA was performed by two operators with skills and long-standing experience in ultrasound-guided venous access in electrophysiology.

### 2.3. Technique for Axillary Vein Access Guided by Ultrasound

In our current practice, USGAVA is the first-choice technique, reserving either axillary venipuncture guided by fluoroscopic landmarks, rarely with venography, or cephalic vein cutdown in cases of unsuccessful USGAVA. The procedure is performed by a single operator using a handheld ultrasound system (Vscan Extend™, GE Healthcare, Waukesha, WI, USA) equipped with a high-frequency 3.3–8 MHz linear array transducer. The detailed technique and results of our unique experience with a pocket-sized handheld ultrasound device for CIED implantation have recently been published [35].

The puncture is performed freehand (i.e., without needle guides) transcutaneously, before making the skin incision. Unlike other operators, we favor this approach to avoid complications related to handling the probe within a tight surgical pocket, as well as to prevent potential compromise of ultrasound image quality due to the presence of micro air bubbles that could enter the tissues surrounding the AV during tissue dissection to create the surgical pocket. To prepare for venipuncture, the ultrasound transducer is tilted to acquire an image of the infraclavicular AV along its longitudinal axis, preferably where it courses over the body of the second rib. The AV is easily distinguishable from the adjacent axillary artery based on specific characteristics such as its superficial position, compressibility under gentle pressure from the probe, lack of pulsation, respiratory collapsibility, and visualization of the typical angled entry of the cephalic vein. Additionally, turbulent flow produced by the saline infusion from the ipsilateral arm may be a reliable marker. Then, we advance an 18-gauge needle while keeping it aligned with the ultrasound beam plane to clearly visualize the entire needle shaft profile and its complete trajectory as it progresses through the tissues. After visualizing the needle tip-caused indentation on the anterior wall of the AV, the needle is gently advanced with short jabs until it enters the lumen. In the first attempt, we direct the needle tip towards the section of the vein running above the body of the second rib. This approach minimizes the risk of puncturing the lung in case the needle accidentally passes through the posterior wall of the vein. After completing AV cannulation, either the wide-open infusion of saline or the Trendelenburg position is discontinued. Following a successful puncture, a 0.035 inch j-tip guidewire is introduced through the needle into the vein and navigated under fluoroscopy to the inferior vena cava. In cases wherein multiple leads are needed, additional venous access points are obtained with the same method, performing an ultrasound-guided puncture for each lead by adjusting the puncture site proximally along the course of the AV by 0.5 cm. Alternatively, the operator may choose to use a single-puncture method with a retained guidewire for implanting multiple lead. Next, a linear skin incision is made medially to the guidewires and a subcutaneous device pocket is created above the pectoralis major muscle fascia. When an electrosurgical unit is used, the guidewire(s) inside the vein are covered with a 5 French dilator introduced transcutaneously, and kept in place as long as the electrical current is delivered to avoid thermal damage to the tissues. Blunt dissecting scissors are used to access the guidewires through the subcutaneous tissue, pulling them beneath the skin and into the device pocket. Subsequently, a peel-away dilator/introducer assembly is introduced over the guidewire into the central venous system, and the leads are implanted following standard procedures.

When USGAVA fails, we create a subcutaneous device pocket, and an alternative venous access is then obtained through either cephalic vein cutdown or AV puncture guided by fluoroscopic landmarks, following standard techniques. When AV puncture is chosen under fluoroscopic guidance, the image of the body of the first rib becomes the reference point for venous puncture.

### 2.4. Routine Maneuvers to Increase Filling of Axillary Vein

Maneuvers or procedures capable of increasing venous filling and, consequently, venous lumen, optimize the likelihood of a successful USGAVA. Therefore, to prepare patients for the procedure, we routinely administer intravenous fluids and use the Trendelenburg position to increase AV diameter and minimize its respiratory collapsibility. Therefore, a few minutes before the ultrasound scanning begins, we initiate a wide-open saline infusion through a peripheral vein in the ipsilateral arm for device implantation. Concomitantly, patients are shifted from a supine to a Trendelenburg position by tilting the operating table head-down at an initial angle of 5°, which can be adjusted up to 15° as needed, based on the AV size and patient tolerance.

### 2.5. Use of the Valsalva Maneuver

The Valsalva maneuver is used in an ad hoc manner when the operator deems the AV unsuitable for puncture due to a narrow diameter (conventionally defined at our center as antero-posterior diameter < 6 mm), or when cyclic collapse of the AV during the inspiration phase of tidal breathing makes optimal ultrasound visualization challenging (Figure 1).

Therefore, the patient was preliminarily instructed to take a deep breath, close the glottis, and attempt to exhale forcefully against the closed airway, including the nostrils. When requested by the operator, the nursing staff encouraged and assisted the patient in performing the Valsalva maneuver and holding their breath in forced exhalation for as long as possible during the venipuncture. According to our experience, a steadily expanded AV is typically achieved within 3–4 s from the initiation of the Valsalva maneuver, allowing an expert operator to complete the venipuncture attempt.

### 2.6. Variables and Definitions

Baseline demographic data, biometric and clinical characteristics, procedural specifics, and complications that occurred within a 30-day post-procedure window were prospectively collected, either as continuous or categorical variables. Images of the AV were acquired and stored within the handheld ultrasound device. Consequently, the captured images underwent manual off-line review to assess the maximum diameters of the AV in the long-axis view, both during tidal breathing and, when performed, the Valsalva maneuver. To assess the reproducibility of the AV measurements, a reliability analysis was conducted as outlined: the same operator reevaluated vein diameter across the first 80 patients undergoing USGAVA to evaluate intraobserver variability; subsequently, an additional independent operator, unaware of prior measurements, assessed the AV diameter in a subset of 20 out of the 80 patients to estimate interobserver variability. The overall procedural duration was measured from the administration of local anesthesia to the completion of the skin suture. A venous access attempt was deemed effective when all the leads designed for implantation were visualized inside the inferior vena cava using fluoroscopy. If any method apart from USGAVA was used, even if only for one lead in instances of multiple-lead devices, the procedure was classified as unsuccessful. A chest X-ray was consistently conducted the day after the procedure, with the patient in an upright position when feasible, to assess the presence of pneumothorax or lead displacement. The skin incision was subjected to daily examination throughout the hospitalization period, and subsequently assessed upon the removal of skin sutures, typically performed around day 12 post-implantation.

### 2.7. Study Endpoints

The study endpoints encompassed the following procedural outcomes: successful venous access and cannulation, overall procedural duration, and cumulative X-ray exposure time. Furthermore, we gathered data regarding complications occurring within one month after CIED implantation, including pneumothorax, pocket hematoma necessitating intervention (e.g., deescalation of antithrombotic therapy, implementation of drainage measures), infection, venous thrombosis of the accessed vein, as well as any other issues associated with the procedure.

### 2.8. Statistical Analyses

Statistical analyses were performed using R software (R version 4.2.1) and RStudio environment (Posit team 2023). Continuous variables are expressed as means ± standard deviations if normally distributed or medians (25th–75th percentile) if not normally distributed. All continuous variables were visually assessed for normality by inspecting the density distribution and formally tested using the Shapiro–Wilk Test. Categorical data are expressed as counts and percentages. Continuous variables were compared using the Wilcoxon rank sum test for independent samples. Categorical variables were compared using a Chi-square test or Fisher’s exact test when appropriate. For the analysis of intraobserver and interobserver AV measurement variability, we used the intraclass correlation coefficient. The effect of the Valsalva maneuver on the axillary vein diameter (longitudinal variation) was assessed using a paired-samples Wilcoxon test. Two-tailed tests were considered statistically significant at the 0.05 level.

## 3. Results

During the study period, we enrolled 148 patients who underwent CIED implantation with USGAVA for lead insertion (Figure 2).

The characteristics of the entire studied population are shown in Table 1.

According to our definition, a successful USGAVA was obtained in 137 patients (92.6%), resulting in 254 out of 273 leads (93%) effectively implanted. Procedure-related complications occurred in two patients (1.8%) who nonetheless underwent a successful USGAVA and CIED implantation. In one patient, deep venous thrombosis was diagnosed in the upper limb, ipsilateral to the implantation site, two weeks after dual-chamber pacemaker implantation. The remaining patient experienced accidental puncture of the axillary artery during the initial USGAVA attempt, with no subsequent consequences.

In 41 cases, the operator asked the patient to perform the Valsalva maneuver to enhance the likelihood of a successful USGAVA procedure. Out of these, four patients were unable to cooperate and execute the required maneuver. As a result, they underwent AV cannulation using alternative methods to USGAVA. Therefore, we collected data from 37 patients who underwent USGAVA while performing the Valsalva maneuver. Baseline characteristics and descriptive data of patients undergoing USGAVA are presented as two separate groups, with and without the Valsalva maneuver, in Table 2.

Briefly, patients who were asked to perform the Valsalva maneuver had a higher body mass index (28.6 ± 4.2 kg/m^2^ vs. 26.3 ± 3.7 kg/m^2^, *p* = 0.007) and body surface area (1.9 ± 0.2 m^2^ vs. 1.8 ± 0.2 m^2^, *p* = 0.009). Additionally, their AV diameter during tidal breathing was considerably narrower (2.7 ± 1.7 mm vs. 9.1 ± 3.3 mm, *p* < 0.0001), and the success rate of USGAVA lower (81% vs. 96%, *p* = 0.006) compared to patients who were not required to perform the Valsalva maneuver. The two groups were otherwise similar in all other variables. Overall, the Valsalva maneuver resulted in the AV diameter increasing from 2.6 (3.4) mm to 8.2 (3.9) mm (*p* < 0.001), with a median gain of 5.6 mm. In more detail, Figure 3 illustrates the absolute change in AV diameter achieved with the Valsalva maneuver for each of the 37 patients who performed it. The Valsalva maneuver doubled the maximum AV diameter in 21 patients (57%), and yielded a value ≥ 6 mm in 28 patients (76%).

In Table 3, the characteristics of patients who performed the Valsalva maneuver are presented as two separate groups based on the success or failure of the USGAVA procedure.

Among these patients, the USGAVA procedure was unsuccessful in seven cases (19%), either due to the operator decision not to proceed with it (*n* = 5) or due to procedure failure (*n* = 2). These patients showed a trend toward a higher body mass index (31.3 ± 4.3 vs. 28.0 ± 4.0, *p* = 0.062) and exhibited a significantly narrower AV diameter during tidal breathing (0.9 ± 0.4 mm vs. 3.2 ± 1.6 mm, *p* = 0.001) compared to the Valsalva patients with a successful USGAVA. Additionally, the Valsalva maneuver led to a notably smaller increase in their AV diameter (0.5 ± 0.3 mm vs. 6.0 ± 2.8 mm, *p* < 0.0001) that resulted in a narrower maximal AV diameter (1.3 ± 0.6 mm vs. 9.2 ± 2.5 mm, *p* < 0.0001) in comparison to the Valsalva patients who achieved successful USGAVA.

Finally, as outlined in the Materials and Methods section, during USGAVA we consistently aim to puncture the AV as it runs over the body of the second rib. As a result, only in 4 out of 107 patients who achieved successful USGAVA without the Valsalva maneuver did the operator opt to perform the venipuncture outside the boundaries of the second rib. In contrast, among patients with effective USGAVA during the Valsalva maneuver, venipuncture occurred more proximally in 20 out of 30 patients (67%), as the AV exited beyond the upper edge of the second rib.

## 4. Discussion

To the best of our knowledge, no other research has been conducted to evaluate the feasibility, effectiveness, and safety of the Valsalva maneuver performed by patients undergoing CIED implantation, as an additional strategy to increase the success rate of USGAVA procedure. The main findings of the study can be summarized as follows: most of the patients (90%) were able to correctly perform the Valsalva maneuver as instructed by the operator; the Valsalva maneuver effectively increased the AV diameter in most patients; USGAVA carried out during the Valsalva maneuver in patients with AV initially deemed unsuitable for venipuncture was highly effective (81%), and no maneuver-related complications, either acute or mid-term, were observed.

In our study, operators used the Valsalva maneuver to facilitate AV puncture of selected patients who underwent USGAVA for endovascular CIED leads placement. Our approach was pragmatic, relying on simple verbal instructions, conveyed by the treating staff in the regular clinical setting for CIED implantation, to describe the Valsalva maneuver without standardizing it to a specific target pressure to achieve or a specific duration for breath-holding. However, since the Valsalva maneuver requires patients’ cooperation and the ability to hold their breath during forced exhalation against closed airways until the needle tip enters the AV, concerns might arise, especially in the case of elderly patients or those with cognitive decline. In our series, despite the advanced age (mean 76 ± 10 years) of the 41 patients who were asked to perform the Valsalva maneuver, the procedure was feasible for most patients, as only 4 of those were found to be unable to do so. Only limited data are available regarding the ability of patients to perform the Valsalva maneuver, and these studies were conducted in different clinical settings, with younger patients, and using different or unspecified definitions of effective Valsalva maneuver. To our knowledge, these results are uniquely available, demonstrating the high feasibility of the Valsalva maneuver, even in an elderly population and within an interventional clinical setting.

The presence of a thin or highly collapsible AV represents key challenging factor for a successful USGAVA. The Trendelenburg position and Valsalva maneuver have been largely investigated with regard to their effect to enhance venous filling, increase venous size, and reduce collapsibility during central venous catheterization through different veins [36,37,38,39], with conflicting results regarding their effects on the AV. Lim et al. reported a significant increase in the cross-sectional area of the extra-thoracic subclavian vein (i.e., the axillary vein), as measured using 2-D ultrasound images, at the end of full expiration compared to end-inspiration, both in the supine position and in the Trendelenburg position at 15°, in a group of 20 healthy young adult volunteers [34]. Hightower and Gooding showed that the ultrasound-assessed anteroposterior diameter of the extra-thoracic subclavian vein increased when performing the Valsalva maneuver, in 11 healthy young adult volunteers [33]. Unlikely previous studies, Ford et al. reported a lack of effect from the Valsalva maneuver performed during either supine or Trendelenburg position on the cross-sectional area of the AV, as measured by ultrasound, in 30 stable emergency young-adult patients [40]. In the present study, the AV size increased significantly (*p* < 0.001) when patients performed the Valsalva maneuver during the Trendelenburg incline, resulting in a median gain in venous antero-posterior diameter of 5.6 mm. Our findings contrast with those of Ford. These differences can, in part, be attributed to the diversity of the study populations, as our study included older patients and used different measures for AV size (longitudinal diameter versus cross-sectional area). Furthermore, our systematic use of wide-open saline infusion may have mitigated the negative effects of hypovolemic status on AV filling. In a study on 80 patients undergoing CIED implantation, we found a positive relationship between successful USGAVA and AV antero-posterior diameter, with a 3-fold increase of probability of success per each 1 mm increase in the AV diameter [41]. Additionally, an AV antero-posterior diameter ≥ 5.7 mm was found to be highly accurate in predicting successful USGAVA, with a sensitivity of 84% and specificity of 100% [41]. In the present study, patients who were asked to perform the Valsalva maneuver exhibited a maximal AV diameter during spontaneous breathing that was notably thin, especially when compared with that of the remaining patients (2.7 ± 1.7 mm vs. 9.1 ± 3.3 mm, *p* < 0.0001). Since operators used the Valsalva maneuver in those patients initially deemed to have inadequate AV for venipuncture, it can be speculated that the Valsalva maneuver, by increasing AV size and counteracting its collapsibility, raised the overall success rate of USGAVA up to 93%. Indeed, in 30 out of 37 patients who were able to perform the Valsalva maneuver, CIED implantation was achieved using USGAVA for leads’ placement. In the case of the remaining seven patients for whom AV was still considered unsuitable for USGAVA during the Valsalva maneuver, leads’ insertion was guided by fluoroscopic landmarks. Furthermore, the Valsalva maneuver did not impact the well-established safety of USGAVA, as no complications, whether acute or observed during the 30-day follow-up, were recorded. The safety of the Valsalva maneuver in this clinical setting is emphasized by the high percentage of patients (58%) receiving antithrombotic therapy while undergoing CIED implantation, particularly concerning the risk of hematoma and hemothorax.

As mentioned earlier, our initial attempt is to puncture the AV over the body of the second rib. Consequently, operators initially assess the suitability of the AV at this level. It is worth noting that this segment of the AV is surrounded by firm structures, including the bony structure of the rib cage itself and the skeletal muscle tone of pectoralis muscles, which may hinder its expansion despite maneuvers aimed at increasing venous filling, regardless of body position and patient age [42]. Furthermore, the AV vein has been reported to be implicated in as much as 3–5% of cases of thoracic outlet syndrome [43]. The subcoracoid space represents the specific region within the thoracic outlet where compression of the AV can occur. This anatomical compartment is bounded anteriorly by the minor pectoral muscle, posteriorly by the second or third rib, and superiorly by the coracoid process. Functionally acquired factors, such as muscle hypertrophy or fibrosis, can lead to constriction within the subcoracoid space, consequently hindering the transvenous access and advancement of cardiac leads, including those used for temporary cardiac pacing [44]. This could explain why operators found it necessary to ask for the Valsalva maneuver from more than a quarter of the assessed patients, most of them exhibiting a thin AV during tidal breathing. Likely for the same reason, we observed that during the Valsalva maneuver, the maximum diameter of the AV was often measured more proximally than at the conventional first attempt venipuncture site, where the axillary vein becomes unobstructed by the posterior bony structure of the second rib. This led operators to puncture the AV as it exits the body of the second rib, thus without the protection provided by its bony shield. Nonetheless, there were no cases of pneumothorax.

Finally, it is interesting to note that the Valsalva maneuver did not affect either the overall procedure duration or the X-ray exposure time, with the same number of catheters to be implanted per patient as those who were not asked to perform the maneuver.

Some limitations need to be addressed in the present study. First, it is a prospective, not randomized, single-center study with a small sample size. Therefore, further larger, multicenter, and randomized studies are necessary to confirm our results. Secondly, since lead revisions and device upgrades have been excluded from the analysis, our findings cannot be extrapolated to include such procedures. Thirdly, since we have not collected data on mid- and long-term complications, we are unable to provide safety data for comparison beyond the 30-day follow-up period. Fourthly, the decision to request the performance of the Valsalva maneuver was partially subjective, considering factors such as the collapsibility behavior of the AV, and partially guided by an AV diameter cut-off value, conventionally defined at our center as 6 mm. However, we exercise caution in recommending this cut-off value as the sole criterion for deciding whether to proceed with Valsalva-facilitated USGAVA, as the supporting evidence is limited, derived from a single-center and nonrandomized study [15]. Fifthly, as our approach aimed to be pragmatic, the Valsalva maneuver was based on standardized verbal instructions without requiring a standardized target pressure and breath-hold duration to be achieved; therefore, we cannot rule out the possibility that some non-responsive patients in the Valsalva group might not have performed the maneuver properly. Finally, since both USGAVA procedures, with or without the Valsalva maneuver, were carried out by highly experienced operators in ultrasound-guided venous access within the field of electrophysiology, it is important to note that our results may not be replicable when performed by operators with less experience.

## 5. Conclusions

To our knowledge, these results represent the first published experience of using the Valsalva maneuver during CIED implantation. In our series, the Valsalva maneuver has proved to be a feasible, effective, and safe maneuver to both increase axillary vein size and facilitate USGAVA in selected patients undergoing a first endovascular pacemaker or cardioverter-defibrillator implantation. Our pragmatic approach, combined with the Trendelenburg position, could potentially enhance the overall success rate of USGAVA, particularly for patients initially considered unsuitable due to smaller vessel size and high collapsibility of the AV.

## Figures and Tables

**Figure 1 diagnostics-13-03274-f001:**
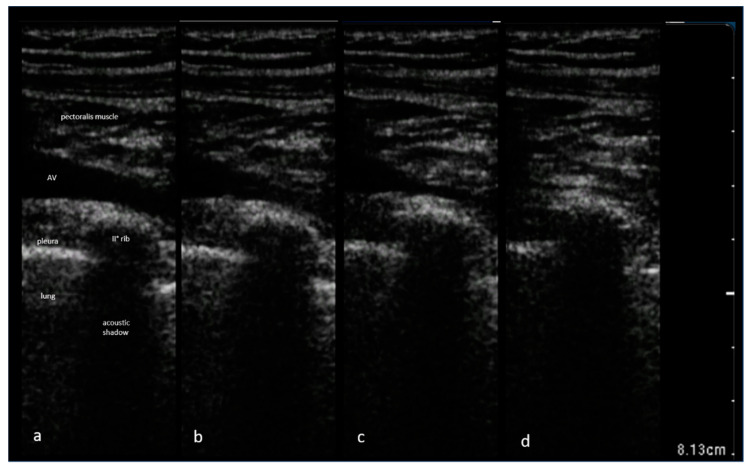
Ultrasound imaging of the axillary vein size variation during tidal breathing at the initial (**a**), mid (**b**,**c**), and final (**d**) inspiratory phase. In frame (**d**), captured at the end of inspiratory act, the axillary vein lumen appears slender, making it unsuitable for ultrasound-guided venipuncture.

**Figure 2 diagnostics-13-03274-f002:**
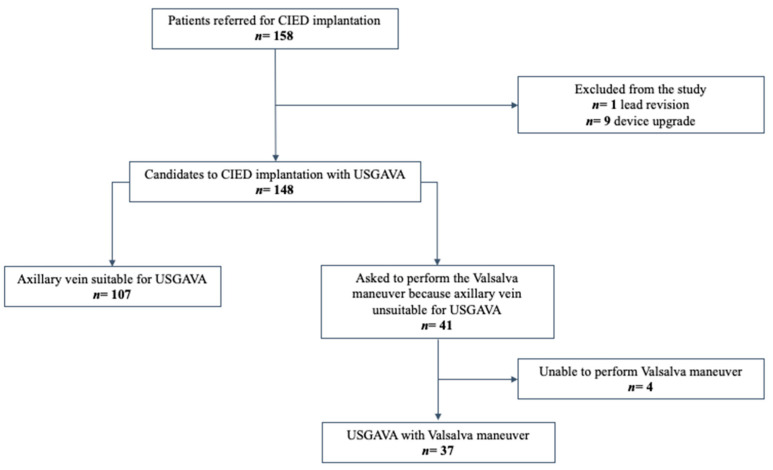
Flowchart depicting the selection and exclusion process of the initial population, and the final study group of patients.

**Figure 3 diagnostics-13-03274-f003:**
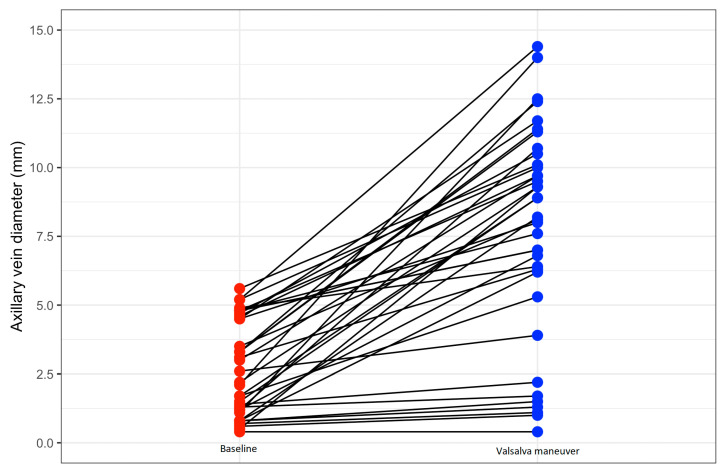
Changes in antero-posterior longitudinal diameter of axillary vein, as assessed by ultrasound, obtained at baseline, during tidal breathing, (red circles) and during the Valsalva maneuver (blue circles), for each of the 37 patients who performed the maneuver.

**Table 1 diagnostics-13-03274-t001:** Demographic, clinical, and procedural characteristics of the study patients.

Characteristic	*n* = 148 ^1^
Age (year)	78 (8)
Male gender	104 (70%)
BMI (kg/m^2^)	26.8 (3.9)
BSA (m^2^)	1.9 (0.2)
LVEF (%)	50.6 (14.5)
Diabetes mellitus	30 (20%)
Coronary artery disease	40 (27%)
Chronic obstructive pulmonary disease	16 (11%)
Hypertension	105 (71%)
History of cardiac surgery	17 (11%)
Creatinine (mg/dL)	1.0 (0.4)
Use of diuretic therapy	45 (30%)
Antithrombotic therapy	
No antithrombotic therapy	35 (24%)
Oral anticoagulant	70 (47%)
Single antiplatelet therapy	36 (24%)
Dual antiplatelet therapy	2 (1.4%)
Oral anticoagulant + antiplatelet therapy	5 (3.4%)
Device type	
Single-chamber pacemaker	29 (20%)
Dual-chamber pacemaker	69 (47%)
Biventricular pacemaker	23 (16%)
Single-chamber cardioverter-defibrillator	13 (8.8%)
Dual-chamber cardioverter-defibrillator	2 (1.4%)
Biventricular cardioverter-defibrillator	12 (8.1%)
Success rate of ultrasound-guided axillary vein access	137 (93%)
Complications	2 (1.4%)
Total number of implanted leads	273 (0)
Average number of leads per patient	1.8 (0.6)
Procedural time (min)	68.1 (32.5)
X-ray time (min)	2.8 (1.6, 6.2)
Axillary vein depth (mm)	21.7 (6.6)
Axillary vein diameter (mm)	7.5 (4.1)

^1^ Mean (SD); *n* (%); Median (IQR).

**Table 2 diagnostics-13-03274-t002:** Comparison between patients undergoing ultrasound-guided axillary vein access, with and without the Valsalva maneuver.

Characteristic	Without Valsalva *n* = 111 ^1^	With Valsalva *n* = 37 ^1^	*p*-Value ^2^
Age (year)	79 (8)	76 (10)	0.075
Male gender	75 (68%)	29 (78%)	0.213
BMI (kg/m^2^)	26.3 (3.7)	28.6 (4.2)	0.007
BSA (m^2^)	1.8 (0.2)	1.9 (0.2)	0.009
LVEF (%)	51.2 (13.4)	48.6 (17.3)	0.545
Diabetes mellitus	22 (20%)	8 (22%)	0.813
Coronary artery disease	30 (27%)	10 (27%)	1.000
Chronic obstructive pulmonary disease	11 (9.9%)	5 (14%)	0.548
Hypertension	82 (74%)	23 (62%)	0.174
History of cardiac surgery	13 (12%)	4 (11%)	1.000
Creatinine (mg/dl)	1.0 (0.4)	1.0 (0.4)	0.372
Use of diuretic therapy	30 (27%)	15 (41%)	0.122
Antithrombotic therapy			0.011
No antithrombotic therapy	19 (17%)	16 (43%)	
Oral anticoagulant	58 (52%)	12 (32%)	
Single antiplatelet therapy	28 (25%)	8 (22%)	
Dual antiplatelet therapy	1 (0.9%)	1 (2.7%)	
Oral anticoagulant + antiplatelet therapy	5 (4.5%)	0 (0%)	
Device type			0.212
Single-chamber pacemaker	24 (22%)	5 (14%)	
Dual-chamber pacemaker	50 (45%)	19 (51%)	
Biventricular pacemaker	20 (18%)	3 (8.1%)	
Single-chamber cardioverter-defibrillator	7 (6.3%)	6 (16%)	
Dual-chamber cardioverter-defibrillator	1 (0.9%)	1 (2.7%)	
Biventricular cardioverter-defibrillator	9 (8.1%)	3 (8.1%)	
Success rate of ultrasound-guided axillary vein access	107 (96%)	30 (81%)	0.006
Complications	2 (1.8%)	0 (0%)	1.000
Average number of leads per patient	1.9 (0.6)	1.8 (0.5)	0.371
Procedural time (min)	70.6 (33.7)	60.4 (27.7)	0.128
X-ray time (min)	3.2 (1.6, 6.2)	2.4 (1.6, 5.5)	0.647
Axillary vein depth (mm)	21.4 (6.8)	22.5 (6.2)	0.356
Axillary vein diameter (mm)	9.1 (3.3)	2.7 (1.7)	<0.0001

^1^ Mean (SD); *n* (%); Median (IQR). ^2^ Wilcoxon rank sum test; Pearson’s Chi-squared test; Fisher’s exact test.

**Table 3 diagnostics-13-03274-t003:** Comparison between patients performing the Valsalva maneuver categorized on the success or failure of the ultrasound-guided axillary vein access procedure.

Characteristic	Unsuccessful *n* = 7 ^1^	Successful *n* = 30 ^1^	*p*-Value ^2^
Age (year)	70 (13)	77 (9)	0.095
Male gender	6 (86%)	23 (77%)	1.000
BMI (kg/m^2^)	31.3 (4.3)	28.0 (4.0)	0.062
BSA (m^2^)	1.9 (0.2)	1.9 (0.1)	0.790
LVEF (%)	48.6 (19.3)	48.6 (17.2)	0.997
Diabetes mellitus	2 (29%)	6 (20%)	0.631
Coronary artery disease	2 (29%)	8 (27%)	1.000
Chronic obstructive pulmonary disease	0 (0%)	5 (17%)	0.560
Hypertension	5 (71%)	18 (60%)	0.687
History of cardiac surgery	0 (0%)	4 (13%)	0.570
Creatinine (mg/dL)	0.8 (0.1)	1.1 (0.4)	0.069
Use of diuretic therapy	3 (43%)	12 (40%)	1.000
Antithrombotic therapy			0.189
No antithrombotic therapy	4 (57%)	12 (40%)	
Oral anticoagulant	1 (14%)	11 (37%)	
Single antiplatelet therapy	1 (14%)	7 (23%)	
Dual antiplatelet therapy	1 (14%)	0 (0%)	
Oral anticoagulant + antiplatelet therapy	0 (0%)	0 (0%)	
Device type			0.355
Single-chamber pacemaker	0 (0%)	5 (17%)	
Dual-chamber pacemaker	5 (71%)	14 (47%)	
Biventricular pacemaker	0 (0%)	3 (10%)	
Single-chamber cardioverter-defibrillator	1 (14%)	5 (17%)	
Dual-chamber cardioverter-defibrillator	1 (14%)	0 (0%)	
Biventricular cardioverter-defibrillator	0 (0%)	3 (10%)	
Complications	0 (0%)	0 (0%)	1.000
Total number of implanted leads	13 (0)	52 (0)	
Average number of leads per patient	1.9 (0.4)	1.7 (0.6)	0.597
Procedural time (min)	62.9 (15.3)	59.8 (30.1)	0.797
X-ray time (min)	2.0 (1.8, 3.3)	2.5 (1.7, 6.2)	0.575
Axillary vein depth (mm)	21.6 (6.1)	22.8 (6.3)	0.659
Axillary vein diameter with tidal breathing (mm)	0.9 (0.4)	3.2 (1.6)	0.001
Axillary vein diameter with Valsalva maneuver (mm)	1.3 (0.6)	9.2 (2.5)	<0.0001
Axillary vein diameter increase with Valsalva maneuver (mm)	0.5 (0.3)	6.0 (2.8)	<0.0001

^1^ Mean (SD); *n* (%); Median (IQR). ^2^ Two-sample *t*-test; Fisher’s exact test.

## Data Availability

The data presented in this study are available on request from the corresponding author.

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
