# Peer review of "Feasibility of Ultrasound-Guided Axillary Vein Puncture under Valsalva Maneuver for Diagnostic and Cardiovascular Interventional Purposes: Pacemaker and Cardioverter-Defibrillator Implantation"

_diagnostics, 2023, doi:10.3390/diagnostics13203274_

Round 1

Reviewer 1 Report

I have received a manuscript entitled „Feasibility of Ultrasound-Guided Axillary Vein Puncture under Valsalva Maneuver for Diagnostic and Cardiovascular Interventional Purposes: Pacemaker and Cardioverter-defibrillator Implantation” by B. Sassone et al. for review.

The authors present their experience in using the Valsalva maneuver during CIED implantation. In their series, the Valsalva maneuver increased the axillary vein size and facilitated ultrasound guided axillary vein puncture for CIED implantation.

I find this study interesting and well conducted. The manuscript is written in a clear and understandable way. The description of the technique is detailed and has an educational value.

Nonetheless, there are some mistakes in the manuscript, that need to be corrected. Please find them listed below:

1. line 51: „...USGAVA avoids...” - I believe it is not gramatically correct

2. line 62 and many more times in the text (line 137, 138, 374): „opportunistic” -  I do not understand that word in that context. Maybe it is some false friend word from Italian, but please explain what is this supposed to mean.

3. line 101 – the procedure is described in present tense, but here it suddenly changes into past; please unify the gramatic forms

4. line 119: „as long as” eletrical current is delivered instead of „until”? – please check the intended meaning of that phrase

5. line 124-126: when USGAVA fails, the cephalic vein cutdown seems reasonable, but is it really possible to puncture the vein using traditional landmarks if USGAVA failed?

6. Figure 1 – frames are labelled a-d in the picture, but there is frame (e) in the figure legend – please correct

7. line 147 – mind the tenses present/past

8. lines 173-177: the same sentence repeated 2 times?

9. lines 190-191: visual inspection of data distribution is not enough for verification of normality, an appropriate statistical test should be used. Moreover, the statistical proof of normality is a prerequisite for specific statistical tests for intergroup comparisons. As the authors report, the Wilcoxon rank sum test was used for some comparisons, but the the t-test was used for repeated measures, and it needs normality of data. Was it correctly used?

10. What is the meaning of table 2? It does not add anything to the manuscript and should be removed.

11. Table 3 – total number of implanted leads is common for both groups, and from the table it appears 273 for each group, that line shold be removed

12. As I understand, Figure 2 and 3 present exactly the same data. In figure 2 values start below zero, which is incorrect. Anyway, figure 3 is more explanatory than figure 2, and I would recommend leaving only Figure 3 (moving it to the results section) and deleting Figure 2.

13. Table 4 – the line „Success rate of ultrasound-guided axillary vein access” makes no sense, as this is the variable, based on which the authors divided the group into two subgroups for the purpose of that table. Please remove that line.

14. line 274 – full expiration or inspiration?

15. lines 373 and 374 – PM and ICD should be singular and not plural (i.e. pacemaker and cardioverter-defibrillator)

as listed above

Reviewer 2 Report

Thank you very much for the opportunity to review this interesting work:

Feasibility of Ultrasound-Guided Axillary Vein Puncture under

Valsalva Maneuver for Diagnostic and Cardiovascular Inter ventional Purposes: Pacemaker and Cardioverter-defibrillator 4 Implantation." The manuscript is interesting and discusses an important topic of interventional cardiology. I suggest creating and adding a Flowchart to have an idea of ​​what large population the material came from. I suggest citing the works DOI: 10.5603/FC.a2022.0034 in the discussion PMID: 31037504; PMCID: PMC6514035. discussing the problem of technical difficulties in the implantation of a temporal electrode due to the thoracic outlet syndrome
